# Sexual Dysfunction in Patients with Chronic Non-Genital Physical Disease: An Umbrella Review

**DOI:** 10.3390/ijerph22020157

**Published:** 2025-01-24

**Authors:** Charlotte Leemans, Stephan Van den Broucke, Céline Jeitani

**Affiliations:** IPSY, 1348 Louvain-la-Neuve, Wallonia, Belgium; stephan.vandenbroucke@uclouvain.be (S.V.d.B.); celine.jeitani@uclouvain.be (C.J.)

**Keywords:** chronic disease, sexual dysfunction, umbrella review

## Abstract

Many primary studies and reviews have been published on the influence of chronic diseases (CDs) on sexual dysfunction (SD), but CDs not involving the genitals are less well represented in the literature. Aim: To determine the prevalence of SD in patients with non-genital CD and assess the overall impact on sexual functioning. Methods: An umbrella review was performed of published systematic reviews on the relationship between the occurrence of CDs not involving the genitals and patients’ sexual functioning, following the PRISMA guidelines. PubMed, EMBASE, the Cochrane Library, PsycInfo, Scopus, and CINAHL were used to find publications for inclusion in the review, with two independent researchers performing the search and selection of articles, data extraction, and quality assessment. The relative risk (RR) or odds ratio (OR) with 95% confidence was used as an indicator of the association between CD and SD across studies. The quality of studies included in the review was assessed using Amstar-2. Outcomes: Forty-three systematic reviews, thirty of which included a meta-analysis, met the inclusion criteria, representing a total of 880,586 patients (756,629 (85.9%) men, 68,388 (7.8%) women, and 55,569 (6.3%) unspecified). Results: Among men, CD is associated with erectile dysfunction, and among females, with lower levels of desire, arousal, lubrication, orgasm, and sexual satisfaction and with increased pain during intercourse. For both men and women, depression, anxiety, and fatigue are also reported, while women with CD are more affected by a poor body image than men. Clinical implications: Patients with CD, especially females, should be more routinely assessed for the impact of their condition on sexual functioning. The impact of CD on men’s sexuality has been extensively studied in terms of erectile capacity, but other aspects of their sexuality are largely neglected. Strengths and limitations: This is the first umbrella review to bring together the documented findings regarding sexual dysfunction among patients with various non-genital CDs. While the heterogeneity of the CDs makes the study unique and clinically relevant, it renders the interpretation of the results more difficult. The overrepresentation of men in existing studies reflects the current state of research but limits the applicability of the findings for women. Conclusions: Women and men with non-genital CDs can suffer from SD or reduced sexual function. Health professionals should pay more attention to managing these sexual disorders, even when the disease does not affect the genitals.

## 1. Introduction

Sexual health refers to “a state of physical, mental and social well-being related to sexuality” [1]. Sexuality is a marker of health: it is important for maintaining good mental health and a basic need that cannot be separated from other aspects of human life. Yet, while quality of life is significantly influenced by sexual functioning [2], the latter can be impaired by CD [3]. It has been widely documented that living with a CD can cause significant changes in the sexual functioning and/or body image of patients, which may affect the quality of their affective and sexual relationships as well as those of their partners. In addition, undergoing treatment or surgery for a chronic disease can bring about changes in one’s physical appearance and in the hormonal, vascular, and neurological systems [4]. These changes can also negatively affect sexual desire and self-esteem, thus reducing the patient’s sexual satisfaction and psychological well-being [5]. This effect can be further exacerbated by psychological factors such as depression, anxiety, and antidepressant use, which can have a significant impact on sexual functioning and affect the quality of their relationship with their partners [6].

Chronic diseases, defined as long-term conditions lasting at least six months and evolving over time, represent a significant global health burden, accounting for approximately 17 million deaths annually [1]. The Community of Patients for Research [7] classifies CDs into 15 categories: (1) cardiovascular diseases, (2) cancers, (3) endocrine diseases, (4) respiratory and ear, nose, and throat (ENT) diseases, (5) digestive system diseases, (6) rheumatological diseases, (7) neurological and muscular diseases, (8) psychiatric and psychological diseases, (9) renal, urinary and genital diseases, (10) skin diseases, (11) eye diseases, (12) systemic diseases, (13) chronic infectious diseases, (14) hematological diseases, and (15) neurodiversities. This classification is also recognized in the list of CDs adapted from the International Classification of Primary Care, drawn up by the International Organization of General Practitioners [8] and the International Classification of Diseases, which provides a common language for CD [9]. Such classifications are critical for understanding the diverse health impacts of chronic diseases, including their influence on sexual functioning and overall quality of life.

Sexual functioning is defined as a person’s ability to experience sexual pleasure and satisfaction when desired [10]. A reduction in sexual functioning can lead to sexual distress. Sexual distress can be broadly described as the inability to enjoy sexual intercourse to the full. The reasons for this inability may be organic or non-organic [11]. Organic sexual dysfunction includes erectile dysfunction (ED), vaginismus, and dyspareunia, whereas non-organic dysfunction includes a lack of sexual desire, sexual aversion, a lack of sexual enjoyment, the failure of genital response, orgasmic dysfunction, premature ejaculation, non-organic vaginismus, and non-organic dyspareunia [12]. The World Health Organization defines sexual dysfunction as “a disturbance of sexual desire and psychophysiological changes that characterize the sexual response cycle and result in marked distress and interpersonal difficulties” [13]. Sexual functioning primarily encompasses the ability to actively participate in satisfying sexual activities, including arousal, desire, and orgasm. It is assessed using various criteria such as the frequency of intercourse, quality of arousal, and overall satisfaction with sexual life. Sexual dysfunction is often diagnosed by means of self-report questionnaires or interviews. It should be noted that sexual functioning should not be confused with sexual distress, which refers more to the negative emotions associated with impaired or diminished sexual functioning. This distress can be caused by relationship difficulties, past traumas, addiction problems, and marital conflicts, as well as mental or physical health problems. It is possible for a person to have good sexual functioning but experience sexual distress, and vice versa. Our study focuses on the impact of chronic diseases on sexual functioning and the resulting sexual distress [11].

Despite the large number of patients who experience sexual dysfunction when suffering from a chronic condition, many clinicians do not consider sexual health issues during consultation [14]. As a result, the impact of the disease on the patient’s sexual functioning is often underestimated and insufficiently addressed [4]. As revealed by a meta-synthesis of qualitative studies [15], this may partly be due to a lack of time during the consultation, but also due to personal reasons, such as a lack of knowledge and awareness amongst health professionals, fear of offending the patient, or embarrassment. Physicians often do not consider the patient’s sexuality to be part of their profession or may feel incompetent with regard to the matter due to a lack of training, as sexology courses are often missing in their curriculum. Others may hesitate to ask their patients about their sexual functioning, considering it to be part of their private life [16].

In the literature on the relationship between CD and sexual disorders, studies involving CD that affect the genitals are much more prevalent than those focusing on diseases that do not involve the sexual organs. Sexuality has long been studied primarily from the perspective of reproduction, with attention focused on the genital organs and reproductive functions. This perspective has led to a reductive view of sexuality, limited to its genital aspects [17]. As such, gynecologists and urologists are more aware of the impact of a disease on sexual health than, for example, gastroenterologists or diabetologists. Also, oncologists are more likely to discuss sexuality when a patient has cancer of the cervix, breast, or testicles than if he or she is affected by lung or colon cancer. Nevertheless, discussing the impact on sexual functioning of any chronic disease is important for the disease’s (self)-management. Since the latter not only involves addressing the disease symptoms and the treatment, but also coping with the chronic nature of the disease, properly addressing the sexual dysfunction that is associated with a disease is beneficial to the patient’s general well-being and disease self-management. In addition, a sexual dysfunction may be an indication of a CD of which the patient is as yet unaware, such as pre-diabetes [18], hyperprolactinemia [19], or heart failure [20]. As such, paying attention to sexual functioning can help to diagnose these hidden CD.

The impact of specific CDs on sexual functioning is well documented; however, an integrative review examining this relationship across a range of CDs is currently lacking. This study seeks to fill this gap by reviewing the existing literature on the prevalence of sexual dysfunction among patients with various chronic non-genital physical illnesses and analyzing the impact of these conditions on patients’ sexual functioning.

## 2. Methods

To address the above question, an umbrella review was performed according to the Preferred Reporting Items for Systematic Reviews and Meta-analysis (PRISMA) guidelines [21]. The protocol for the review was registered on PROSPERO under ID: CRD42022306975. As this study focused on published research articles, it was not necessary to obtain ethical approval or informed consent from participants.

### 2.1. Search Strategy

A systematic search was performed using PubMed, CINAHL Complete, Embase, PsycInfo, Cochrane Library, and Scopus between 8 and 24 June 2022. A second search was performed in March 2023 and December 2023 to include the most recent publications. The equations for the search terms (available on request) were developed around four basic themes: chronic diseases, sexual problems, adults, and literature review. The keywords for the search were the MeSH terms and thesauri considered appropriate for the sexual dysfunctions under study and for the chronic diseases that are listed in the ComPaRe platform, with the addition of free vocabulary (Table 1). For the Cochrane Library database, only the MeSH terms were included and no free vocabulary, as the number of terms was limited, and the original query string was too long for the software. No language criterion was applied, and no restrictions were placed on the publication date of the articles.

### 2.2. Eligibility Criteria

To be included in the review, publications had to meet the following inclusion criteria: (1) be a systematic review with or without a meta-analysis focusing on adult humans; (2) involve patients with a CD (i.e., cardiovascular diseases; cancers; endocrine, respiratory, or ENT diseases; diseases of the digestive system; rheumatological, neurological, muscular, kidney, skin, eye, systemic, chronic infectious, or hematological diseases; chronic fatigue; and chronic pain); and (3) investigate the impact of the chronic disease on sexual functioning by considering one of the following sexual dysfunctions: sexual desire disorders, sexual arousal disorders (including erectile dysfunction), orgasm disorders (anorgasmia and anejaculation), premature ejaculation, or sexual pain (dyspareunia and vaginismus).

The following publications were not included (exclusion criteria): (1) Publications focusing on chronic genital gynecological disorders such as endometriosis, uterine fibroids, ovarian cysts, and fertility disorders, or on chronic genital urological disorders such as benign prostatic hypertrophy and adenoma, bladder leakage, overactive bladder, or primary erectile dysfunction. Also excluded were studies related to cancers of the genitals such as bladder, prostate, breast, cervical, vaginal, penile, and testicular. As these have an obvious impact on sexual functioning, they have already been studied extensively, and as such do not concur with the focus and objective of the present review. Although the breast is not really a sexual organ, studies on the effects of breast cancer on sexual functioning were also excluded for the same reason. Conversely, studies on anal cancer were included, as the anus is not considered a genital area. Other conditions excluded from the study were paraphilic disorders and gender identity disorders, as well as chronic malformations occurring after surgery (which are no longer listed as disorders), and chronic mental disorders, the management of which is very different from that of chronic physical diseases.

The sexual repercussions of aging and menopause were also excluded, as these are not considered chronic diseases. Aging and menopause are natural physiological processes and part of the normal course of life. While they can significantly impact sexuality, they do not meet the criteria for chronic diseases, which typically involve an underlying pathology or medical disorder requiring specific management.

Depression and anxiety are not considered as chronic diseases included in the study, because they are either chronic mental illnesses, which fall under the exclusion criteria, or comorbidities of other CDs.

In terms of populations, studies on children or adolescents were excluded from the review, as were studies involving animal testing because the links with sexuality cannot be established in any meaningful way in these contexts. (3) For the purposes of this umbrella review, narrative reviews, case studies, primary reviews, clinical trials, cohort studies, case-control studies, and randomized controlled trials were excluded. (4) In addition, reviews not published in peer-reviewed journals, such as doctoral theses and conference presentations or posters, were also excluded.

#### Study Selection

Applying the criteria outlined in the previous section, a search of review studies was performed using the Cadima Software [22], which is an evidence synthesis tool for systematic reviews that automatically identifies and removes duplicates. A first selection was made on the basis of the title/abstract, followed by a full text evaluation of the selected articles. When an article was not immediately available, authors were contacted by email to obtain the full version and/or data. To secure intersubjectivity, the selection of the publications for inclusion was performed by two independent researchers (CL, CJ), whereby the criteria were fine-tuned in a preparatory phase until sufficient interrater reliability was achieved. Any disagreements were resolved by consensus or by consultation with a third person (SV).

The literature search yielded 3997 potentially relevant records, of which 2592 were retained after removal of duplicates. After screening of the titles and abstracts, 77 articles were considered for full-text evaluation, of which 35 unique studies met the criteria for inclusion in the review. The update of 2023 yielded eight additional review studies, bringing the total to 43 published review studies. Reasons for exclusion from studies can be provided on request. The PRISMA flowchart detailing the search and selection process is given in Figure 1.

Of these 43 studies, 30 involved a meta-analysis and 13 a systematic review without meta-analysis. Together, they involved a total of 88,0586 patients with non-genital CD, 756,629 (85.9%) of whom were male, 68,388 (7.8%) of whom were female, and 55,569 (6.3%) whose gender was not specified. Twelve of the studies were concerned with women only [23,24,25,26,27,28,29,30,31,32,33,34], 14 with men only [35,36,37,38,39,40,41,42,43,44,45,46,47,48], and 17 with both male and female patients [49,50,51,52,53,54,55,56,57,58,59,60,61,62,63,64,65,66]. In terms of diseases, 2 of the 43 studies involved patients suffering from cardiovascular diseases (stroke [59], hypertension [32]), 4 involved cancers (colon [54], young adult cancer [62], low-grade glioma [63], Hodgkin’s lymphoma [35]) 8 involved endocrine diseases (diabetes [33,36,41], hypothyroidism [58], Klinefelter syndrome [37], polycystic ovarian syndrome [26,27,31]); 3 involved respiratory tract diseases (COPD [47,49], obstructive sleep apnoea [57]); 2 involved gastroenterological diseases (inflammatory bowel disease [52], cirrhosis [43]); 7 involved rheumatological diseases (musculoskeletal pains [50], fibromyalgia [24], rheumatoid arthritis [29,48,66], gout [39], and systemic autoimmune rheumatic diseases [34]); 3 involved neurological and muscular diseases (Parkinson [55], multiple sclerosis [30,61]); 3 involved nephrological diseases (chronic renal failure [53], end-stage renal disease [28,40]); 4 involved dermatological diseases (psoriasis [46], vitiligo [65], hidradenitis suppurativa [60,64]); 2 involved systemic diseases (lupus [56], primary Sjögren’s syndrome [23]); 1 involved inflammatory periodontal disease [38]; 3 involved hematological diseases (hyperuricemia [44,45], acute leukemia [51]); and 1 involved chronic pain (vulvodynia [25]). Eleven studies specifically investigated the impact of the disease on male erectile dysfunction.

### 2.3. Data Extraction

The full-text versions of all publications meeting the inclusion criteria were reviewed by two independent researchers (CL, CJ). They extracted the following information:Study characteristics: authors, year of publication, title, study design, number of reviewers, databases consulted, keywords, inclusion and exclusion criteria, quality assessment, and diagnostic tool used for identifying sexual dysfunction;Population characteristics: gender, type of chronic disease, country of the study, and sample size;Clinical findings: presence of sexual dysfunction, counseling provided, study limitations, and main conclusions.

The data extracted from the included studies were summarized in a Microsoft Excel spreadsheet. Any disagreements or discrepancies were resolved through consensus or, if necessary, by consulting a third investigator (SV).

### 2.4. Quality Assessment

The quality of the studies selected for the review was assessed independently by two assessors (CL and CJ) using the AMSTAR-2 tool [67]. This tool allows for an evaluation of the quality of systematic reviews on 16 criteria: (1) formulation of the research question; (2) provision of an a priori design; (3) explanation of the choice of the study design of the studies included in the review; (4) comprehensiveness of the literature search; (5) adequacy of the study selection; (6) adequacy of the data extraction; (7) availability of a list of excluded studies, along with reason for exclusion; (8) comprehensiveness of the description of the main features of the studies included; (9) risk of bias assessment; (10) inclusion of information about the sources of funding for the studies included in the review; (11) methods for statistical combination of results; (12) assessment of the potential impact of risk of bias of individual studies on the meta-analysis result; (13) discussion/interpretation of the potential impact of risk of bias of individual studies on the meta-analysis result; (14) discussion of the heterogeneity observed in the study results; (15) consideration of the likelihood of publication bias; and (16) declaration of study authors’ conflict of interest. Each of these criteria can be checked using the response options “yes”, “partially”, or “no”. Seven of the criteria (i.e., 2, 4, 7, 9, 11, 13, and 15) can affect the validity of the review and its conclusion and are therefore considered “critical domains”. While AMSTAR 2 is not intended to give a quantitative assessment of the review studies’ quality, it provides a useful set of criteria to detect potential weaknesses. As such, it allows one to distinguish between “high quality” studies (which show no or only one non-critical weakness); “moderate quality” studies (showing more than one non-critical weakness but no critical flaws); “low quality” studies (showing one critical flaw with or without non-critical weaknesses); and “critically low quality” studies (showing more than one critical flaw with or without non-critical weaknesses).

## 3. Results

### 3.1. Study Quality

The evaluation of the quality of the systematic reviews included in this umbrella review revealed that nineteen of them could be regarded as “high quality”, ten as “moderate quality”, nine as “low quality”, and five as “critically low quality” (information on the quality of each article is available as Appendix A). The most common critical shortcomings were failure to provide a list of excluded studies with the reasons for exclusion (absent in six studies, and only partially provided in seventeen studies), the absence of an a priori design (only partially achieved in ten studies), insufficient comprehensiveness of the literature search (only partially achieved in ten studies), and poor estimation of bias or risk assessment (absent in nine studies). For studies involving a meta-analysis, an assessment of the potential impact of the risk of bias of individual studies on the results of the meta-analysis was absent in 9 out of 27. Other, less critical flaws were the failure to carry out a selection of studies or to perform data extraction in duplicate (seven and six studies, respectively), a lack of comprehensiveness of the description of the main features of the studies (only partially achieved in twelve studies), a lack of discussion of the heterogeneity observed in the studies, and insufficient consideration of the likelihood of publication bias (not addressed in three studies). None of the studies provided information about the sources of funding for the studies included in the reviews.

### 3.2. Assessment of Sexual Functioning

As shown from the review studies included in this umbrella review, the most frequently used tools to assess sexual functioning are the International Index of Erectile Function (IIEF) [68,69] for males and the Female Sexual Function Index (FSFI) [70] for females. The IIEF is a 15-item questionnaire which measures sexual functioning in men across five subscales (erectile function, orgasmic function, sexual desire, sexual satisfaction, overall satisfaction). A total score ranging between 5 and 25 (for IIEF-15) can be calculated as the sum of the scores on the five subdimensions, enabling a diagnosis of severe (5–7), moderate (8–11), mild to moderate (12–16), mild (17–21), and no (22–25) erectile disorder. The FSFI is a 19-item self-report scale that assesses women’s sexual functioning over a 4-week period, on the basis of six criteria: (1) desire (the motivation to engage in sexual activity); (2) orgasm (the ability to reach orgasm); (3) lubrication (the ease or difficulty with which lubrication is achieved and maintained during sexual activity); (4) arousal (the ability to become cognitively aroused); (5) satisfaction (the satisfaction with one’s actual sexual experiences); and (6) pain (feeling genital pain during or after vaginal penetration). With a maximum score of six on each subscale, the maximum total score is 36, whereby a higher score indicates a better sexual functioning. A score of less than 26 is considered an indication of a clinically low level of sexual function.

### 3.3. Impact of Chronic Disease on Sexual Functioning

The details from the analysis of the 12 review studies that are concerned with women are summarized in Table 2, those of the 17 studies involving men are presented in Table 3, and those of the 17 concerning both male and female patients are presented in Table 4.

#### 3.3.1. Cardiovascular Disease

As documented by a review by Dusenbury [59] involving 1701 male and 182 female patients, the prevalence of SD after stroke ranges from 29% to 94.8%. Although a return to sexual activity usually occurs between the third and sixth month after a stroke, continued sexual dissatisfaction is common, and the decrease in sexual frequency is significant in 44% to 77% of the patients. After a stroke, sexual problems may persist for another 2 years. Women who had a stroke are significantly affected by changes in vaginal lubrication and sexual inhibition, which affect both sexual desire and orgasmic function. Among men who experienced a stroke, 60% report dysfunctions regarding ejaculation (whereby it is not specified whether it concerns premature ejaculation or anejaculation). Control groups have not always been clearly defined in all studies and the quality was assessed as “critically low”, so the results must be interpreted cautiously. An additional stressor is that of reliving a stroke during sexual activity. This underscores the fact that although stroke is considered an acute illness, it often has chronic effects.

A review of studies of FSD in 1057 hypertensive women [32] revealed a prevalence of FSD of 14–90% in normotensive women. The study also highlighted that hypertensive women were 1.81 times more likely to develop FSD than non-hypertensive women [32].

#### 3.3.2. Cancer

A review by Stanton and colleagues [62] involving 1416 young male and 873 female cancer patients revealed that adolescents and young adults with cancer suffer from alterations in sexual function, sexual desire, orgasm, and sexual satisfaction. Young men additionally report problems with erection and ejaculation, whereas young women report impaired arousal, lubrication, and increased pain during sex. Control groups are not always clearly defined in all studies and the quality was assessed as “critically low”, so the results must be interpreted with caution.

A review of studies of SD in 2777 male patients with colon cancer [54] showed a prevalence of ED of 44 to 78% compared to 10 to 15% in the general population. The prevalence of ejaculation problems was 22 to 47%, compared to 16 to 23% in the general population. In women, the prevalence of vaginal dryness among colon cancer patients is 28%, compared to 14–19% in the general population, and that of dyspareunia is up to 27%, compared to 13% in the general population. Male and female colon cancer patients report pain upon penetration in 9 to 27% of cases (compared to three to five percent in the general population). Control groups were not considered in this systematic review, and gender was not specified.

A recent comprehensive review of the sexual health of 517 people with hematological malignancies, particularly acute leukemia [51], reported that 29 to 35% of them reported decreased sexual activity, while 9 to 18% experienced difficulties with sexual arousal, including erectile dysfunction in men and vaginal dryness or dyspareunia in women. Of 163 patients with chronic myeloid leukemia, 44% to 62% reported decreased sexual activity, and 38% to 55% reported decreased sexual desire. The same review concluded that among patients with Hodgkin’s or non-Hodgkin’s lymphoma (*n* = 1484 and *n* = 2145, respectively), sexual dysfunctions were very common, with up to 63% of patients reporting decreased sexual activity and up to 73% decreased sexual desire. This concurs with Arden-Close et al.’s [35] review study of 1710 men with Hodgkin’s lymphoma, showing that the prevalence of SD in this group of patients varies from 20 to 54%. Problems with orgasm were reported by 5 to 58% of both male and female patients with Hodgkin and non-Hodgkin lymphoma. Compared to other cancers, Hodgkin’s appears to have a greater impact on sexual function than other leukemias, but a lower impact than testicular cancer. It is noted that no review studies have been undertaken on the impact of Hodgkin’s lymphoma on the sexual functioning of women. Control groups were not considered in this systematic review, and gender was not specified.

#### 3.3.3. Endocrine Diseases

Endocrine diseases have an effect on SD in both men and women. Nevertheless, the majority of studies on SD among patients with endocrine diseases have focused on male erectile dysfunction (ED). A systematic review of studies of men with diabetes involving a total of 88,577 patients [36] reported a prevalence of ED of 59.1% (95% CI = 55.5 to 62.7), or more than one in two patients. A similar prevalence of ED of 54.3% (95% CI = 28.2–80.5) was reported in a meta-analysis of 2003 diabetic men living in Ethiopia [41]. The prevalence of ED is significantly higher among men with type 2 diabetes than among those with type 1 (66.3% versus 37.5%, *p* < 0.0001), and among those who are over 60 years of age (66.7%). Men with diabetes have a 3.62 times higher risk of developing an ED than men without diabetes. Moreover, men with diabetes tend to develop erectile dysfunction 10 to 15 years earlier than non-diabetics [41], and patients who are over 40 years of age are 4.42 times more likely to develop ED than those who are younger (OR = 4.42, 95% CI = 2.83–6.00) [41]. The duration of the disease also influences the risk of ED, in the sense that patients who have diabetes for more than 5 years have a 3.2 times higher risk of developing ED (OR = 3.2, 95% CI = 1.74–4.66).

A review of 3168 women with diabetes concluded that these women have a higher risk of developing SD than women without the disease (type 1 = OR 2.27 (95% CI = 1.34–4.168; *p* = 0.0002) and type 2 = OR 2,49 (95% CI =1.55–3.99; *p* = 0.178)) [71]. The total FSFI score is also lower in women with any type of diabetes, who are between 1.6 and 3.3 times more likely to have low FSFI scores. The study also highlights that the FSFI score is almost twice as likely to decrease if the subject has a BMI above 24 [33].

Other endocrine diseases can also cause SD. A review by Barbonetti et al. [37] involving 608 men with Klinefelter’s syndrome found an overall prevalence of 28% of SD (95% CI =19–36%). The overall prevalence of libido loss among men suffering from the disease was 51% (95% CI = 36–66%), indicating that one in every two men with KS suffer from a decrease in libido. Shen [58] reported that both men and women with hypothyroidism have a 2.26-fold higher risk of SD than the general population (RR = 2.26, 95% CI = 1.42 to 3.62, *p* < 0.001). For women, the relative risk of developing SD is even slightly higher (RR = 2.39, 95% CI = 1.31–4.39, *p* < 0.005).

A review of 6256 women with polycystic ovarian syndrome (PCOS) concluded that these women have a 1.32 times higher risk of developing SD than women without the disease [26]. This concurs with the results of other meta-analyses involving women with PCOS [27,31], which also report lower FSFI scores and risks of SD of up to 1.46 times higher than among healthy women, with particularly higher risks of having low levels of arousal (OR = 1.34), orgasm (OR = 1.36), and lubrication (OR = 1.31). Although the prevalence of 35% of having SD is not significantly higher than the 29% found in women without PCOS, when comparing FSFI scores, women with PCOS have more sexual dissatisfaction.

#### 3.3.4. Respiratory Tract Diseases

Men with chronic obstructive pulmonary disease (COPD) have a 2.89 times higher risk of developing ED than men without COPD (OR = 2.89, 95% CI = 1.93–4.32, *p* < 0.001) [47]. A review of studies involving 1446 COPD patients estimated the weighted prevalence of ED among these men to be 74.6% [57]. Of the men, 28.7% had severe ED, 22.5% moderate ED, 19.5% mild ED, and 22.6% no ED. This estimation confirms the results of a previous review with a meta-analysis on a total of 1187 COPD patients, based largely on the same studies, in which the pooled prevalence of ED was 74% (95% CI = 68–80%), compared with 56% (CI = 37–73%) in non-COPD controls [49].

Other respiratory diseases may also impact sexual health. A review by Steinke [57] on 1446 men and 226 women with obstructive sleep apnea (OSA) showed a prevalence of ED among the men ranging from 40.9% to 66.1%, and a prevalence of FSD varying from 32.2% to 71% for women, depending on the selected studies.

#### 3.3.5. Gastroenterological Diseases

A systematic review by Zhang et al. [52] on 2003 men and 34,673 women with irritable bowel disease (IBD) indicated SD prevalence of 27% among men (95% CI 25–29%, *p* < 0.001), while women with IBD (*n* = 34,673) had a prevalence of SD of 53% (95% CI 50–55%, *p* < 0.001). Among male patients with an active IBD, the risk of developing SD increases by a factor of 2.73. Treatment for IBD also has an influence: patients requiring a treatment with corticosteroids have 2.62 times the risk of developing SD; those who undergo surgery for IBD have a risk that is 1.33 times higher; those who are treated with antihypertensive drugs have a 2.60 risk increase; and those who receive biological medication have a risk that is 5.81 times higher [52]. This meta-analysis did not provide information on control groups. A meta-analysis of 770 men with liver cirrhosis [43] showed a prevalence of ED of 79.08% (95% CI = 68.00–88.42). The prevalence of severe ED was 33% (95% CI = 23–44), of moderate ED was 10% (95% CI = 4–17), of mild to moderate ED was 13% (95% CI = 4–25), and of mild ED was 11% (95% CI = 6–17). As liver function deteriorates, the prevalence of ED increases, with the prevalence of ED among compensated cirrhosis patients being 53.61% (95% CI = 35.95–70.84) and among decompensated cirrhosis patients being 88.39% (95% CI = 77.64–96.32). The proportion of patients with severe ED among decompensated cirrhosis is 35% compared to 18% in compensated cirrhosis. This meta-analysis does not provide information on the control group, nor does it specify the sample size according to the type of cirrhosis.

#### 3.3.6. Rheumatological Diseases

A review study by Perez-Garcia et al. [48] found that among 6625 men with rheumatoid arthritis, the risk of developing SD was 2.7 times higher than among men without the condition (OR = 2.7, 95% CI = 1.09–6.05), with SD prevalences ranging from 33 to 62%. Among men with antiphospholipid syndrome, the prevalence of SD ranged from 42 to 45.5%, and among 842 male patients with spondyloarthropathy, it ranged between 30 and 82.5%. Luo et al. [39] reported that the 5011 men with gout the included in their review have 1.2 times the risk of developing ED (RR = 1.20, 95% CI = 1.10–1.31, *p* < 0.001) than men without the disease, whereas Wang et al. [44] in their review of 6083 patients with hyperuricemia found the risk of ED to be 1.59 times higher (OR = 1.59, 95% CI = 1.29–1.97). A meta-analysis involving 85,406 men with hyperuricemia [45] found a prevalence of ED of up to 33%, which rose up to 50% if the patient also had type 2 diabetes as a comorbidity, but fell to 4% when diabetic patients were excluded. It may be that hyperuricemia and ED may simply share common risk factors related to a metabolic syndrome, without necessarily having a causal link. This study provides no information on the control groups. Among women, Zhang [29] reported poorer sexual functioning in a sample of 346 women with rheumatoid arthritis (RA). RA patients scored lower on each FSFI dimension, particularly on the lubrication dimension. The low score for women with polyarthritis on lubrication may be due to treatment with glucocorticoids, which can lead to decreased or abnormal androgen levels. Moreover, the average age of the women in the five studies included in Zhang et al. [29]’s meta-analysis was 43 years (ranging between 36.68 and 48.65), which is close to the menopausal period. Inadequate estrogen levels often lead to sexual dysfunction in women due to adverse effects such as vaginal atrophy, increased vaginal pH levels, and genital tract infections, which may explain the decrease in vaginal lubrication. The impact of RA on sexual functioning among women is confirmed by another meta-analysis [66], demonstrating that the 306 women involved showed significantly lower values for desire (OR = 8.35), arousal (OR = 6.36), lubrication (OR = 6.36), orgasm (OR = 10.76), satisfaction (OR = 9.48), and pain (OR = 3.97), as well as total FSFI scores (OR = 8.05) compared to healthy controls, particularly for the orgasm dimension.

#### 3.3.7. Neurological and Muscular Diseases

A review of studies involving 245 men with systemic sclerosis [48] gave prevalence estimates of SD ranging from 66 to 100%, and of ED ranging from 38 to 100%. A review by Zhao et al. [55] revealed that male patients with Parkinson’s disease have a 1.79 times increased risk of suffering from SD (RR = 1.79; 95% CI = 1.26–2.54, *p* < 0.001). This is higher than for women with Parkinson’s, who have a 1.3 times higher risk of having SD (RR = 1.3, 95% CI = 0.64–2.61, *p* = 0.469). Hypersexuality is not uncommon in Parkinson’s disease patients receiving dopamine replacement therapy [72], yet this meta-analysis did not distinguish between men and women.

A review involving a total of 251 male patients with multiple sclerosis [61] reported a prevalence of SD ranging from 76.9 to 81.4% for this disease, and IIEF scores below 22, which clearly demonstrates that erectile quality is affected. For women with multiple sclerosis, prevalence estimates of FSD have been reported ranging from 46.7% to 86.6% [61]. Their mean FSFI scores are 15.27 (out of a maximum of 36), compared to 25.63 for women without the disease. The areas that appear to have the lowest scores are lubrication, orgasm, arousal, and pain. This systematic review by Gao et al. did not provide any additional information on control groups and was judged to be of critical quality. Another meta-analysis involving 826 women with MS [30] found these women to have a 1.87 times higher risk of developing FSD compared to healthy subjects (95% CI = 1.25–2.78, *p* = 0.002) and to have lower FSFI scores (SMD = 2.41, 95% CI = 3.87–0.96, *p* = 0.017). The FSFI score is also influenced by the duration of the disease, with women with a long duration of MS (>10 years) reporting lower total FSFI scores (SMD = 5.43, 95% CI = 6.48–4.37, *p* < 0.001) than those with a duration of MS of less than ten years (SMD = 1.49, 95% CI = 2.61–0.36, *p* = 0.009). Women with MS have significantly lower values in all areas of female sexual functioning than healthy controls. They are 4.35 times more likely to have a low FSFI total score. For the different domains, they are more likely to have a low score for sexual desire (OR = 2.43), arousal (OR = 3.13), lubrication (OR = 4.76), orgasm (OR = 4.51), satisfaction (OR = 3.7), and pain (OR = 4.35). Among women over 40 years of age, the relationship between MS and FSD is confirmed, which is not the case among women under 40 years of age. These results suggest that as MS patients age, their sexual function declines [30].

With regard to muscular diseases, a review involving 2342 men and women experiencing musculoskeletal pain [50] showed the average IIEF score for the men to be 51.58 on a scale ranging from 5 to 75, signifying erectile dysfunction among individuals with this medical condition. Women with musculoskeletal pain have a total FSFI score of less than 26.55 on a scale ranging from 2 to 36, and 96% of women with non-specific hip arthritis report sexual dysfunction. The systematic review by Briggs et al. does not provide sufficient information on the control groups.

#### 3.3.8. Renal Diseases

A review by Luo et al. [53] of studies involving 325 men with chronic renal failure showed that these patients are 2.95 times more likely than others to develop ED. The total sample of 737 patients presented in this meta-analysis does not specify the percentage of men and women. Although the study distinguishes according to the type of treatment, it is not precise with regard to patient gender. A meta-analysis of studies involving 3490 male chronic renal failure patients treated with hemodialysis reported a prevalence of ED of 79% (95% CI = 75–82%). For those who received a kidney transplant, an ED prevalence of 59% was reported (95% CI = 53–64%); for those treated with peritoneal dialysis, the reported prevalence was 71% (95% CI = 58–83%); and for those with end-stage renal disease, the prevalence was 82% (95% CI = 75–88%). Overall, the total prevalence of ED for these men, regardless of treatment, was 71% (95% CI = 67–74%). Among those with end-stage renal disease (ESRD), the overall prevalence of non-severe ED was 46% (95% CI = 43–50%), and of severe ED, the prevalence was 22% (95% CI = 18–27%). It should be noted that ED is very common among all ESRD patients, especially in those who are starting dialysis [40].

Women with chronic kidney disease receiving treatment have a 2.07 times greater risk of developing SD [28], with FSD being more common in patients receiving peritoneal dialysis (RR = 2.33, 95% CI = 1.29–4.21, *p* = 0.000) and hemodialysis (RR = 2.23, 95% CI = 1.52–3.28, *p* = 0.000) than in patients receiving renal transplants (RR = 1.80, 95% CI = 1.04–3.12, *p* = 0.000). A review of studies involving 2340 women with end-stage renal disease [28] reported a prevalence of SD of 74% (95% CI = 67–80%). Among women with kidney transplantation, the prevalence of SD was 63% (95% CI = 43–81%); among those treated with hemodialysis or peritoneal dialysis, the reported SD was 80% (95% CI = 72–87%) and 67% (95% CI = 46–84%), respectively. Women with end-stage renal disease treated with peritoneal dialysis also have a lower FSFI score.

#### 3.3.9. Dermatological Diseases

A review of studies involving a total of 36,242 men with psoriasis [46] concluded that these patients are 1.35 times more likely to develop erectile dysfunction than healthy persons, and that their IIEF-5 score is significantly lower than in control groups. A meta-analysis of 42,729 men and women with hidradenitis suppurativa (HS) revealed a prevalence of 52 to 60% of ED among male patients and of 51 to 62% SD among women [60]. The probability of developing SD was 38% higher in patients with HS, controlling for age, sex, and depressive and anxiety disorders (OR = 1.38, 95% CI = 1.28–1.48, *p* < 0.001). Men and women with HS also report significantly lower IIEF scores [60,64]. Patients with HS have greater sexual distress than healthy controls, and females more so than males. Another study [65] revealed that women with vitiligo score significantly lower on the AVFSFI (Arabic version of the FSFI scale) than control groups, with a mean difference of 0.73 (95% CI = 1.49–0.03; *p* = 0.001). Statistical differences are observed for desire (MD = 0.26; 95% CI = 0.48–0.03; *p* = 0.03), satisfaction (MD = 1.11; 95% CI = −1.49–0.73; *p* < 0.00001), and lubrication (MD = −0.47; 95% CI = −0.91–0.03; *p* = 0.04). The gender distinction was not systematically specified in the selected studies, which considerably complicates an accurate impact assessment. What is more, the sample size was limited to just 155 people, which raises questions about the representativeness of the survey.

#### 3.3.10. Systemic Disease

A review by Perez-Garcia et al. [48] reported that among 234 men with lupus, the prevalence of developing SD varied from 12 to 68%. Women with lupus are 1.56 times more likely to have SD than women without the disease and have a lower overall FSFI score [56]. Women with Sjogren’s syndrome are 5.4 times more likely to have a lower total FSFI score than those without, 3.69 times more likely to report lower sexual desire, 6.96 times more likely to achieve and maintain lubrication during sexual activity, 5.4 times more likely to have low arousal, 2.96 times more likely to have difficulties orgasming and less likely to report sexual satisfaction, and 5.3 times more likely to feel pain [23].

#### 3.3.11. Other Chronic Diseases

For other CDs (excluding psychiatric and psychological diseases), such as eye diseases, neuro-diversities, and hematological or chronic infectious diseases, very few review studies were retrieved. One exception was a review involving 40,773 men with chronic periodontitis, which concluded that these patients had a 1.53 to 3.35 times greater risk of ED than men without this inflammatory disease [38]. Other review studies mentioned the likelihood of SD in patients suffering from hematological or infectious diseases together with other patient groups. For instance, in their review of studies on SD in male and female patients with Hodgkin and non-Hodgkin lymphoma, [51] reported that no problems in this area were found for patients suffering from other hematological diseases.

### 3.4. Addressing Sexual Problems Associated with Chronic Disease

The demonstrated impact of CD on sexual functioning and the moderating role of the disease duration and treatment, the patient’s age, and the presence of comorbidity suggest that more attention should be paid to potential sexual problems in the treatment of CD. One way to achieve this is to improve diagnosis and follow-up by using validated screening tools such as the IIEF to detect sexual problems, ensuring early referral to specialists, and involving the specialist physician more in addressing sexual health issues [48]. In addition, pharmacotherapy can be offered to treat erectile dysfunction [53,57], for example, by prescribing urate-lowering treatment in men with hyperuricemia to reduce the risk of erectile dysfunction [44] or estrogen replacement in kidney disease [53]. Hormone therapy can also normalize sexual desire, satisfaction, and pain in women with hypothyroidism [58], as prolactin levels are likely to be higher in patients with hypothyroidism, and hyperprolactinemia can lead to decreased sexual desire and erectile dysfunction.

Other possibilities are the use of a continuous positive airway pressure (CPAP) machine for men with sleep apnea, which has also been shown to significantly improve erectile dysfunction and reduce problems with desire, intimacy, arousal, and orgasm [57], as well as increasing physical activity [53] or psychotherapy [66,73]. A personalized sexual rehabilitation intervention is recommended for stroke patients [59].

In general, it seems essential to take a less medical look at sexuality and instead follow a more holistic biopsychosocial approach focusing on chronic patients’ subjective complaints or distress and also include the partner, who is often forgotten, in the treatment [49,54]. This implies communicating and providing information about sexuality during consultations. It has been documented that more than 60% of COPD patients ask to discuss sexual issues, but this is often not done [74]. It is noted that interpersonal barriers, social stigma, and misconceptions about sexuality, such as the idea that “older people with chronic illnesses automatically lose interest in sexuality”, can hinder the initiation of sexual health screening and communication [75].

### 3.5. Comparative Analysis

Due to the heterogeneity of the studies involved in the reviews, it was not possible to perform an overall meta-analysis or compare between diseases. However, the odds ratios from eight studies that assessed the risk of erectile dysfunction using the FSFI could be synthesized, and the odds ratio for low FSFI scores from seven studies could be pooled. When odds ratios were not clearly provided, they were calculated from the standardized mean difference (SMD) available in the base study. The results of these subgroup studies are presented in Table 5.

The data show that diabetes is the CD with the greatest impact on erectile quality, with men suffering from diabetes being almost four times as likely to develop erectile dysfunction than non-diabetic men.

Furthermore, from his meta-analysis, Pyrigis [28] concluded that the type of treatment for kidney disease (i.e., by dialysis or hemodialysis) may have a different impact on the risk of obtaining a low FSFI score. As such, treatment of the disease should be taken into account when assessing the risk of SD.

As for sexual satisfaction in women, fibromyalgia emerges as the CD with the greatest risk of obtaining a low score on the FSFI. It should be noted that the calculated values are quite high, due in particular to Besiroglu’s meta-analysis [24], in which one of the included studies clearly stands out from the others. As this one study may explain the surprisingly high results in terms of low score risk, it is recommended to interpret this finding with caution.

## 4. Discussion

While many studies and reviews have documented the impact of CD on sexual functioning, none of the existing reviews thus far have considered this relationship across different non-genital CDs. To address this issue, the present study aimed to provide a descriptive quantitative summary of the impact of different CDs on sexual functioning, based on published systematic reviews and meta-analyses. To that effect, when possible, prevalence rates and odds ratios were used to calculate the risk of having sexual dysfunction in relation to a chronic non-genital physical disease. The results show that for nearly all of the CDs considered, an increased prevalence or risk of experiencing sexual difficulties was found.

### 4.1. Sexual Problems Related to Chronic Disease

Of the sexual problems reported in relation to CD, erectile dysfunction in men is the most often mentioned. This is not surprising, given that erectile dysfunction is the most often studied sexual pathology in general. Yet, it is interesting to note that of the 43 studies included in this umbrella review, 14 looked exclusively at the impact of CD on men’s sexual health, and seven of these looked specifically at erectile quality. Of the different diseases considered, diabetes reportedly has the strongest impact on erectile function.

While the impact of CD on men’s sexual functioning is mostly focused on erectile dysfunction, the impact on women’s sexuality is considered more broadly. Since the effects of a CD on women are most often measured by means of the FSFI, most studies (and reviews) report effects on all six of the domains that are measured by the instrument, namely sexual desire, lubrication, arousal, pain, orgasm, and satisfaction. Of the CDs considered for this review, fibromyalgia appears to have the strongest impact.

The results of this review highlight a gender disparity, with women being under-represented in studies of the impact of chronic diseases on sexuality. This under-representation has led to a downplaying of women’s health issues, including their sexual health, which is considered a lower priority in studies of human sexuality [76,77]. On the other hand, female sexuality is typically perceived as more complex or difficult to assess, particularly when CD is accompanied by depression or mood disorders, which may discourage researchers from investigating this area in depth [78]. This contributes to a lower understanding of the interactions between chronic diseases and sexuality in women, despite the importance of these issues for their quality of life and overall well-being. Further research is needed to understand why the impact of chronic illnesses on sexuality differs between men and women, and to investigate the sexuality-related factors that influence this relationship and moderate its effects on sexual functioning.

The effects on sexual functioning of cancer, lupus, or hyperuricemia appear to vary between individuals and the treatments received. As such, the reviews summarized in this paper suggest that in addition to the disease proper, medical treatments and associated comorbidities may also influence the sexual functioning of patients.

In terms of treatment for cancer, the effect on women’s sexuality depends on the type of surgery. Being able to preserve one’s ovaries, for example, or the possibility of avoiding a mastectomy will decrease the impact on sexual functioning. On the other hand, treatment-induced amenorrhea has been mentioned as an important predictor of sexual dysfunction in women [63]. Similarly, the type of treatment of kidney disease, i.e., by kidney transplantation, hemodialysis, or perineal dialysis, has a differential impact on sexual functioning, with perineal dialysis reportedly having the lowest effect [28,40]. Furthermore, the uses of medication, particularly antidepressants, to treat pain also contribute to sexual dysfunction. Antidepressants are indeed known to cause treatment-emergent sexual dysfunction in a variable proportion of patients depending on the agent used [73].

Comorbidities such as fatigue, pain, anxiety, and depression can amplify sexual dysfunction. CD is often associated with fatigue and pain [48], whereas anxiety and depression are often related to both a CD itself and to sexual dysfunction. As such, there is often a bidirectional link between these pathologies. For example, patients with Sjögren’s syndrome are more likely to develop anxiety and depression than other patients in the healthy control group [23]. In a similar vein, the cardinal symptoms of pain and fatigue associated with fibromyalgia can have a detrimental effect on the sexuality of female patients, as demonstrated by the significantly lower FSFI scores of women with this condition. Depression and anxiety, which frequently accompany fibromyalgia, can further exacerbate its impact on sexual dysfunction [24]. Mood disorders, particularly anxiety and depression, emerge as significant comorbidities that intensify sexual dysfunction. In gastrointestinal disorders, stress and anxiety induced by the underlying pathology play a critical role in the development of sexual dysfunction, emphasizing the psychological burden these conditions impose on sexual health [79]. Similarly, in men with obstructive sleep apnea syndrome (OSAS), chronic fatigue and sleep disturbances directly affect both mental and sexual health, worsening the symptoms of erectile dysfunction [80].

Some reviews also highlight the impact of age as an exacerbating factor in the sense that the older the patients are at diagnosis, the greater the impact of the disease on their sexual function [53]. For example, patients who develop erectile dysfunction as a result of cirrhosis are on average 5.8 years older than those who do not [42]. For cancers, the effects of the disease on sexual functioning are even more severe with older age and also last for several years after the diagnosis [62]. In women with multiple sclerosis, a relationship with SDF was reported for those over 40 years of age but could not be demonstrated among those under 40. These results suggest that as MS patients get older, their sexual functioning declines [29]. In contrast, for rheumatological diseases, younger patients appear to be at greater risk of developing sexual dysfunction than older ones. Male patients with rheumatoid arthritis who are older than 45 years have a 1.47 times higher risk of developing erectile dysfunction compared to healthy persons, while those under 45 have a 3.30 times higher risk [65]. These results are surprising and may be due to stress disorders in young people or influenced by drug or tobacco use, which are more common in young people. In men with hyperuricemia, a significant risk for the development of ED has been reported among those younger than 60 years old, whereas no such association is found among those who are over 60 years. In this case, it can be seen that advanced age does not contribute to the increase in ED.

The duration of the disease is also a factor that can influence sexual functioning. Women who suffer from MS for more than 10 years have lower FSFI scores [29]. Similarly, male patients who have diabetes for more than 5 years reportedly have a 3.2 times higher risk of developing erectile dysfunction than those with diabetes for less than 5 years [40].

It is also worth noting that obesity can impact sexual function in both sexes, whether through hormonal disorders, blood circulation problems, medical complications, or psychological problems linked to body image and self-esteem. Research has not only established a correlation between obesity and reduced sexual function but has also shown that weight loss can improve sexual function [42]. Other factors that can moderate the effect of CD like cardiovascular disease or diabetes on sexual function are the neurovascular mechanisms underlying genital arousal, which may have a negative impact on sexual function in both sexes [80].

An additional factor moderating the effects of chronic illness on sexuality is patient acceptance and adaptation. Therapeutic patient education holds great promise for stabilizing the disease and controlling its negative impacts. Health literacy, and more specifically, sexual health literacy, are innovative concepts that deserve further consideration in future research. In the case of heart failure, for example, it has been shown that the greater the acceptance of the disease, the higher the quality of every aspect of one’s sexual life, and the lower the incidence of erectile dysfunction [75].

### 4.2. Strengths and Limitations

This study is the first umbrella review to systematically review and synthesize existing systematic reviews on sexual health in people with chronic non-genital physical illness. By adhering to the agreed-upon guidelines and criteria for umbrella reviews, including pre-registration, methodological quality assessment, and the involvement of two reviewers to ensure the intersubjectivity of the review and reduce the risk of bias in selection and reporting, we believe that the study adds to the knowledge base that is important for the treatment of CD.

An umbrella has the advantage of offering a comprehensive synthesis of the available literature, yet compiling the results of several systematic reviews entails risks, notably those of carrying over or exaggerating the errors of the studies that are included in the integrated reviews. The statistical, clinical, and methodological variability of these reviews can jeopardize the reliability of the conclusions drawn. Consequently, careful and critical reading is essential to interpret results correctly and avoid being influenced by possible biases. We are therefore aware that the present study has its limitations.

Firstly, as with any systematic review and meta-analysis, substantial heterogeneity was observed between the studies that were included, in terms of study design, frequency and severity of the disease, duration of the disease, age at diagnosis, sample size, mean age of participants, and participant characteristics. The use of different questionnaires, some of which are non-standardized, to measure and diagnose sexual dysfunction also contributed to the heterogeneity, which makes it difficult to synthesize results.

Secondly, it would have been interesting to perform a meta-analysis of the meta-analyses included in this review, but this was not possible because the inclusion and exclusion criteria of the different studies and the questionnaires used to measure the impact on sexuality were too different.

Thirdly, the instruments that were used for most of the studies included in this review (i.e., the FSFI and the IIEF) have some limitations. While the FSFI is the most widely used questionnaire to measure female sexual functioning, it should be realized that in order to calculate a total sexual functioning score, respondents must be sexually active and have attempted vaginal penetration within the past 4 weeks. However, sexuality should not be measured solely by vaginal penetration. In a similar vein, the IIEF also has some limitations, as it focuses only on current sexual functioning and only considers erectile function, and as such does not take other aspects of male sexuality into account, such as sexual desire and orgasm [68]. Finally, it should be noted that the IIEF and FSFI remain measures of self-reported sexual dysfunction. The variability of the results may therefore still vary according to the patient’s personal interpretations and be influenced by false beliefs or social desirability.

Fourthly, it should be noted that some of the studies included in the review rely on the same database [27,31] so that the patients included in these two studies may be duplicated. This may overrepresent the results of these studies in our conclusions. Furthermore, more men are represented in the pooled sample than women. This may have led to a bias that erectile dysfunction is the main consequence of CD on sexuality, while other and less often studied dysfunctions may be underrepresented.

Fifthly, in several of the studies included in this umbrella review, information on the age of participants and the effect of this variable was not always provided. This can influence the conclusions drawn from the studies. Also, not all selected studies systematically provided essential data, such as gender distinctions or references to control groups. In addition, the use of terms such as “sexual disorder” or “female sexual disorder” remains vague and does not fully capture the concrete impact of the condition, even when odds ratios are provided. It is important to recognize these limitations and take them into account when interpreting our results.

Finally, it is important to note that many of the studies included in this review did not include control groups, which limits the ability to determine whether the reported effects differ significantly from natural trends in sexual functioning in the general population, particularly the increasing incidence of sexual problems with age. Consequently, the conclusions should be interpreted with some caution. In addition, chronic illnesses, including multimorbidity, are particularly common, especially in elderly populations, making it difficult to create perfectly comparable control groups. These methodological limitations must be taken into account and underline the need for future research incorporating more rigorous approaches to explore these interactions more precisely.

### 4.3. Perspectives for the Future

While this umbrella review did not focus on gender, one of its main findings was the imbalance in research on the sexual health of chronic patients, which tends to focus more on male than female patients. This highlights the urgent need for further research into the impact of non-genital CD on women’s sexual functioning, to improve prevention, diagnosis, and treatment strategies and promote gender equality in healthcare.

Additionally, it is crucial to consider the influence of societal and cultural factors, such as the stigmatization of sexuality, religious norms, lack of sex education, and age-related stereotypes, which often hinder the recognition and management of sexual dysfunction in patients with chronic illnesses.

In this context, it seems that doctors could play a more active role by systematically including sexual health in the clinical assessment of chronic illnesses. Similarly, encouraging patients to discuss these issues more openly could lead to better care. A two-way approach, combining greater responsibility on the part of carers and greater involvement on the part of patients, could thus help to improve the management of sexual dysfunction.

Another key point is whether the decline in sexual function associated with CD is primarily due to the direct effects of the disease or the psychological consequences it causes. This distinction warrants further research to better understand the underlying mechanisms and develop targeted interventions to improve sexual health in patients with chronic conditions.

## 5. Conclusions

Sexual function is an important aspect of health, and it is essential that it is better addressed in health care. To the best of our knowledge, this is the first umbrella review to analyze the impact of non-genitalized physical CD on sexual function. The results of the review demonstrate that CD has an important impact on the sexual health of both men and women, and that this impact is moderated by the patient’s age, the duration of the disease, medical treatments, and associated comorbidities. Despite the known effects of CD on sexuality, sexual health remains a frequently neglected aspect of the quality of life of patients with CD. It is recommended that physicians pay more attention to sexual health management, both for CDs involving the genitals and for CDs not involving the genitals.

## Figures and Tables

**Figure 1 ijerph-22-00157-f001:**
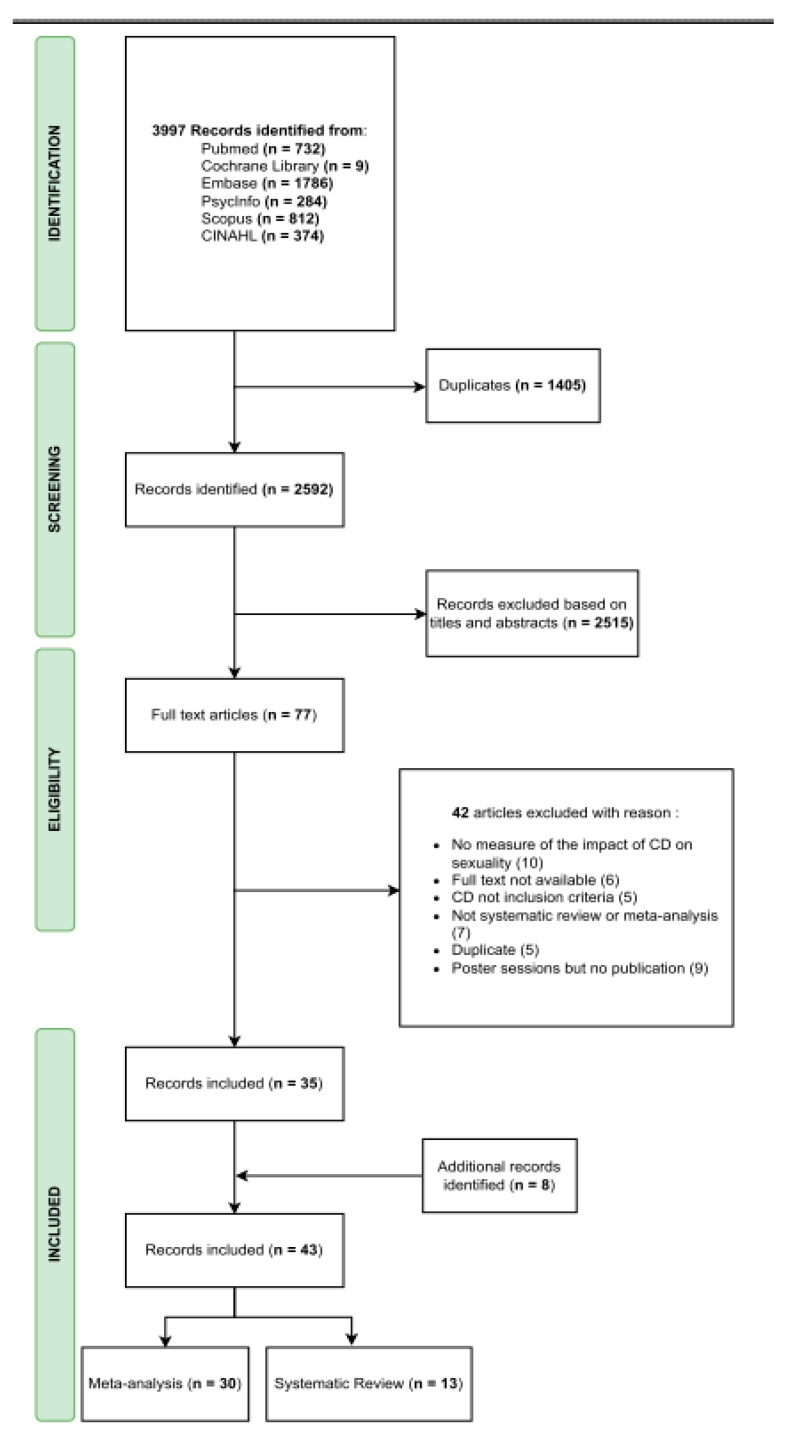
Flow diagram of study selection process.

**Table 1 ijerph-22-00157-t001:** List of search themes and terms used for the search strategy.

Search Themes	Search Terms
Chronic disease	“chronic disease”, “chronic illness”, “cardiovascular diseases”, “fatigue syndrome, chronic”, “chronic pain”, “digestive disease”, “endocrine disease”, “hematologic disease”, “neoplasm”, “neurologic disease”, “respiratory tract disease”, “eye disease”, “skin disease”, “rheumatic disease”, “systemic disease”, “renal disease”.
Sexual problems	“sexual dysfunction, physiological, “sexual dysfunction, psychological”, “sexuality”, “premature ejaculation”, “sexual desire”, “sexual arousal”, “erectile dysfunction”, “libido”, “orgasm”, “vaginismus”, “dyspareunia”
Study design	Systematic reviews
Study population	Adult patients

**Table 2 ijerph-22-00157-t002:** Summary of studies concerning women included in the review.

Study	Type of Study	Disease	N Studies	Patient Group	Control Group	RR or OR	*p*	Prevalence of Sexual Dysfunction	Negative Impacts on Sexuality	Study Quality
Endocrine diseases										
[27]	SR and MA	Polycystic ovary syndrome	18	1932	1987	OR = 1.6	*p* = 0.015	/	Sexual Satisfaction	Moderate
						OR = 1.34	*p* = 0.019		Arousal	
						OR = 1.31	*p* = 0.023		Lubrification	
						OR = 1.36	*p* = 0.028		Orgasm	
						OR = 5.7	*p* < 0.001		Sexual Satisfaction	
[31]	SR and MA	Polycystic ovary syndrome		1163	1463	RR = 1.09; 95% CI 0.9–1.32	/		Female Sexual Disorder	High
[26]	SA and MA	Polycystic ovary syndrome	28	6256	1942	OR 1.32; 95% CI 1.07–1.61	/	35%	Female Sexual Disorder	Low
[58]	SR and MA	Hypothyroidism	6	381	1154	RR = 2.39; 95% CI 1.31–4.39	*p* = 0.005	/	Female Sexual Disorder	Low
[33]	SR and MA	Diabetes type 1	26	3168	2823	OR 2.27 95% CI 1.34–4.168	*p* = 0.002		Female Sexual Disorder	High
		Diabetes type 2				OR 2.49 95% CI 1.55–3.99	*p* = 0.178		Female Sexual Disorder	
		"Any diabetes"				OR 2.02 95% CI 1.49–2.72	*p* = 0.001		Female Sexual Disorder	
Gastroenterological diseases										
[52]	SR and MA	Inflammatory bowel disease	13	2003	/	/	*p* < 0.001	53%: 95% CI (50–55)	Female Sexual Disorder	High
Systemic diseases										
[23]	MA and SR	Primary Sjögren’s Syndrome	3	102	99	OR 5.402	*p* < 0.00001	/	Femal sexual function index	High
Pain diseases										
[25]	SR	Vulvodynia	21	1592	71	/	/	/	Dyspareunia	Moderate
Systemic diseases										
[56]	MA and SR	Lupus		357	564	RR: 1.56 95% CI (0.99–2.48)	*p* = 0.057	/	Sexual Dysfunction	High
Nephrological diseases										
[28]	SR and MA	End–stage Renal Disease	47	2340	/	/	/	74%: 95% CI (67–80)	Female Sexual Disorder	High
		Kidney transplant patients		353	/	/	/	63%: 95% CI (43–81)	Female Sexual Disorder	
		Hemodyalisis		1717	/	/	/	80%: 95% CI (72–87)	Female Sexual Disorder	
		Peritoneal dialysis		270	/	/	/	67%: 95% CI (46–84)	Female Sexual Disorder	
Rheumatology diseases										
[29]	SR and MA	Rheumatoid arthritis	5	346	237	OR: 9,193,307.64	*p* < 0.00001	79.67%: 95% CI (68.75–93.27)	Score Femal sexual function index and Female Sexual Disorder	High
[24]	SR and MA	Fibromyalgia	6	578	341	OR: 9002.6	*p* < 0.0001	/	Femal sexual function index	Moderate
[34]	SR and MA	systemic autoimmune rheumatic diseases	68	5457	/	/	/	63%: 95% CI (56–69)	Female Sexual Disorder	Moderate
[66]	SR and MA	Rheumatoid arthritis	6	273	116	RR 1.73; 95% CI 1.36–2.22	*p* < 0.001	/	Female Sexual Disorder	Moderate
[50]	SR	Musculoskeletal pain	45	4298	/		/	95%	Sexual Dysfunction	Moderate
Nephrological diseases										
[28]	SR and MA	End–stage Renal Disease	47	2340	/	/	/	74%: 95% CI (67–80)	Female Sexual Disorder	High
		Kidney transplant patients		353	/	/	/	63%: 95% CI (43–81)	Female Sexual Disorder	
		Hemodyalisis		1717	/	/	/	80%: 95% CI (72–87)	Female Sexual Disorder	
		Peritoneal dialysis		270	/	/	/	67%: 95% CI (46–84)	Female Sexual Disorder	
Neurological diseases										
[30]	SR and MA	Multiple sclerosis	9	826	659	RR = 1.87; (95% CI 1.25–2.78)	/	/	Female Sexual Disorder	High
						OR = 79.143	*p* = 0.017	/	Femal sexual function index	
[61]	SR	Multiple sclerosis	9	1129	323	/	/	46.7% to 86.6%	Female Sexual Disorder	Critically Low
Oncological diseases										
[62]	SR	Cancer (young adults)	10	873	/	/	/	/	Sexual Frequency, Arousal, Orgasm, Sexual Satisfaction, Dyspareunia	Critically Low
Respiratory diseases										
[57]	SR	Obstructive Sleep apnea	6	226	/	/	/	32.2 % to 71%	Female Sexual Disorder	Moderate
Cardiac Diseases										
[32]	SR and MA	Hypertension	11	1057	715	RR = 1.81 (95% CI 1.10–2.97)	*p* < 0.05	14.1 to 90.1%	Female sexual dysfunction	High
[59]	SR	Stroke		182	/	/	/	29% to 94.8%	Sexual Dysfunction	Critically Low

SR: Systematic Review; MA: Meta-analysis; OR: Odds Ratio; RR: Relative risk; CI: Confidence Interval.

**Table 3 ijerph-22-00157-t003:** Summary of studies concerning men included in the review.

Study	Type of Study	Disease	N Studies	Patient Group	Control Group		RR or OR of Sexual Dysfunction	*p*	Prevalence of Sexual Dysfunction	Negative Impacts on Sexuality	Study Quality
Endocrine diseases											
[36]	SR and MA	Diabetes	145	88,577	/		OR = 3.62; 95% CI 2.53–5.16	*p* < 0.0001	52.5%: 95% CI (48.8–56.2)	Erectile Dysfunction	Low
		Type 1		/	/				37.50%	Erectile Dysfunction	
		Type 2		/	/				66.30%	Erectile Dysfunction	
[37]	SR and MA	Klinefelter Syndrome	16	608	/		/		28%: 95% CI (19−36)	Erectile Dysfunction	High
									51%: 95% CI (36−66)	Decreased Libido	
[41]	SR and MA	Diabete	6	2003	/		/		54.3%: 95% CI (28.2–80.5)	Erectile Dysfunction	Low
[58]	SR and MA	Hypothyroidism	3	79			RR = 2.26; 95% CI 1.42–3.62	*p* = 0.001		Sexual Dysfunction	Low
Dermatological diseases											
[46]	MA	Psoriasis	9	36,242	1657.711		OR 1.35; 95% CI (1.29–1.41)	*p* < 0.00001	/	Erectile Dysfunction	High
Inflammatory diseases											
[38]	SR	Periodontal	4	40,773	685.098		1.53 to 3.35	/	/	Erectyle Dysfunction	Low
Hematological diseases											
[45]	MA	Hyperuricaemia	8	85,406	/	/		33%: 95% CI (13–52)	Erectile Dysfunction	High	
[44]	MA and SR	Hypereuricaemia	11	6083	350.075	OR = 1.59; 95% CI 1.29–1.97	*p* < 0.01	/	Erectile Dysfunction	High	
Oncological diseases											
[35]	SR	Hodgkin’s lymphoma	6	1710	/	/		20% to 54%	Sexual Dysfunction	Critically Low	
[62]	SR	Cancer (young adults)	15	1416	/	/	/	/	Sexual Frequency	Critically Low	
Gastroenterological diseases											
[43]	MA and SR	Cirrhosis	14	770	/	/	/	79.08%: 95% CI (68.00–88.42)	Erectyle Dysfunction	High	
		Cirrhosis decompensated	/	/	/	/	/	88.39%: 95% CI (77.64–96.32)	Erectyle Dysfunction		
		Cirrhosis compensated	/	/	/	/	/	53.61%: 95% CI (35.95–70.84)	Erectyle Dysfunction		
[52]		Inflammatory Bowel Disease	15	34,673	/			27: 95% CI (25–29); *p* < 0.001	Masculine Sexual Disorder	High	
Neurological diseases											
[61]	SR	Multiple sclerosis	3	251	/	/	/	76.9% to 81.4 %	Masculine Sexual Disorder	Critically Low	
Respiratory diseases											
[49]	SR and MA	Chronic obstructive pulmonary disease	12	1187				74%: 95% CI (68–80)	Erectile Dysfunction	High	
[47]		Chronic obstructive pulmonary disease	4	519		OR: 2.89 95% CI (1.93–4.32)	*p* < 0.001	/	Erectile Dysfunction	High	
[57]	SR	Obstructive Sleep apnea	15	1446		/		74.60%	Erectile Dysfunction	Moderate	
Systemic diseases											
[56]	MA and SR	Lupus		123	194	RR: 2.98; 95% CI (2.41–3.68)	*p* = 0.000		Sexual Dysfunction	High	
Cardiac diseases											
[59]	SR	Stroke		1701	/	/	**/**	26% to 60%	Erectyle Dysfunction	Critically Low	
								60%	Ejaculatory Disorder		
Rheumatology diseases											
[39]	SR and MA	Gout	8	5011	14.958	RR 1.20; 95% CI 1.10–1.31	*p* < 0.001	/	Erectile Dysfunction	High	
Perez–Garcia et al. (2020) [48]	SR	Rheumatic diseases	41	8133	39.245	OR 2.7; 95% CI 1.09–6.05		/	Sexual Dysfunction		
		Rheumatoid arthritis	8	6625	38.320	OR 1.67 95% CI 1.36–2.05		33% to 62%	Erectyle Dysfunction	Low	
									Sexual Dysfunction		
		Lupus erythematosus	4	234	175	/		12% to 68%	Sexual Dysfunction		
								68%	Erectyle Dysfunction		
		Antiphospholipid Syndrome	2	23	42	/		42% to 45.5%	Sexual Dysfunction		
								25% to 36.4%	Erectyle Dysfunction		
		Spondyloarthropathies	15	842	525	/		30% to 82.5%	Sexual Dysfunction		
		Systemic Sclerosis	7	245	55	/		66% to 100%	Sexual Dysfunction		
								38% to 100%	Erectyle Dysfunction		
		Behcet Syndrome	5	164	128	/		63% to 80 %	Sexual Dysfunction		
[66]	SR and MA	Rheumatoid arthritis	1	130	391	RR 1.99; 95% CI 1.64–2.43	*p* < 0.001	73% to 99%	Sexual Dysfunction	Moderate	
[50]	SR	Musculoskeletal pain	7	2342	/			72%	Sexual Dysfunction	Moderate	
Nephrological diseases											
[40]	SR and MA	End–stage Renal Disease	94	10,320	/	/		71%: 95% CI (67–74)	Erectile Dysfunction	High	
		Kidney transplant patients	2478	/	/	/		59%: 95% CI (53–64)	Erectile Dysfunction		
		Hemodyalisis	/	/	/	/		79%: 95% CI (75–82)	Erectile Dysfunction		
		Peritoneal dialysis	/	/	/	/		71%: 95% CI (58–83)	Erectile Dysfunction		
		End–stage Renal Disease	/	/	/	/		82%: 95% CI (75–88)	Erectile Dysfunction		

SR: Systematic Review; MA: Meta-analysis; OR: Odds Ratio; RR: Relative risk; CI: Confidence Interval.

**Table 4 ijerph-22-00157-t004:** Summary of studies concerning both men and women included in the review.

Study	Type of Study	Disease	N Studies	Patient Group	Control Group	RR or OR of Sexual Dysfunction	*p* Value	Prevalence of Sexual Dysfunction	Negative Impacts on Sexuality	Study Quality
Dermatological diseases										
[60]	SR	Hidradenitis suppurativa	11	42,729	/	/	/	51% to 62%	Female Sexual Disorder	Critically Low
					/	/	/	52% to 60%	Erectyle Dysfunction	
[64]	SR and MA	Hidradenitis suppurativa	9	1529	/	/	/	/	Score of Female sexual fonction index	Low
						/	/	/	Score of International Index of Erectile Function	
[65]	SR	Vitiligo	4	155	55	/	/	/	Sexual Dysfunction	Low
Nephrological diseases										
[53]	SR and MA	Chronic Renal Failure	10	737	2.988	RR 2.07; 95% CI 1.47–2.91	*p* = 0.000	/	Female Sexual Disorder	High
						RR 2.95; 95 CI 2.16–4.02	*p* = 0.000		Erectile Dysfunction	
Neurological diseases										
[55]	SR and MA	Parkinson’s disease	11	30,150	/	RR = 1.79; 95% CI 1.26–2.54	*p* = 0.001	/	Masculine Sexual Disorder	Moderate
						RR= 1.3; 95% CI 0.64–2.61	*p* = 0.469	/	Female Sexual Disorder	
Oncological diseases										
[54]	SR	Colon	14	2777	/	/	/	24% to 68 %	Sexual Dysfunction	Moderate
					/		/	44% to 78.2%	Erectyle Dysfunction	
					/		/	22% to 47%	Ejaculatory Disorder	
					/		/	28%	Vaginal Dryness	
					/		/	9% to 27 %	Dyspareunia	
[63]	SR and MA	Low grade Glioma	3	124	/	/	/	34% to 47%	Sexual Dysfunction	Moderate
Hematological diseases										
[51]	SR		24	10,506	/	/	/	18% to 50%	Sexual Dysfunction	Low
		Accute leukemia		517	/	/	/	29% to 55%	Sexual Activity	
				206			/	9% to 18%	Arousal	
				629			/	18% to 32 %	Sexual Satisfaction	
				629			/	20% to 24%	DSH	
				878			/	17% to 28%	Other	
		chronic myeloid leukemia		163	/	/	/	44% to 62%	Sexual Activity	
				163			/	34% to 51%	Sexual Satisfaction	
				163			/	38% to 55%	Hypoactive sexual desire	
				163			/	41% to 45%	Other	
		Hodgkin lymphoma		1484	/	/	/	0% to 63%	Sexual Activity	
		non–Hodgkin lymphoma		2145			/	5% to 63 %	Arousal	
				2190			/	12% to 58%	Sexual Satisfaction	
				3476			/	0% to 73%	Hypoactive sexual desire	
				1078			/	5% to 58%	Orgasm	
				5596			/	12% to 50%	Other	

SR: Systematic Review; MA: Meta-analysis; OR: Odds Ratio; RR: Relative risk; CI: Confidence Interval.

**Table 5 ijerph-22-00157-t005:** Summary of risk of low FSFI score and chance of developing erectile dysfunction.

Risks of Erectile Dysfunction (Odds Ratio)	Diabetes [36]	Chronic Periodontitis [38]	Gout [39]	Rheumatoid Arthritis [48]	Psoriasis [46]	Chronic Renal Failure [40]	Chronic Obstructive Pulmonary Disease [47]	Hyperuricemia [45]
N	88,577	40,773	5011	6625	36,242	3490	519	6083
OR	3.62	1.53 to 3.35	1.20	1.67	1.35	2.95	2.89	1.59
Risk of having a low FSFI score (Odds ratio)	Fybromyalgia [24]	Sjogren’s Syndrome [23]	PCOS [27]	Rheumatoid arthritis [66]	Multiple sclerosis [30]	Chronic renal disease with peritoneal dialysis [28]	Chronic renal disease with hemodialysis [28]	
N	578	102	1932	306	826	134	169	
Desire	2227	3.69	NS	8.35	2.43	18.88	22.23	
Arousal	9940	5.4	1.34	6.36	3.13	12.44	9.8	
Lubrification	12	6.96	1.31	6.36	4.76	8.66	8.9	
Orgasm	32,046	2.96	1.36	10.76	4.51	16.94	19.58	
Pain	49	5.3	NS	3.97	4.35	/	4.67	
Satisfaction	9000	2.96	NS	9.48	3.7	10.96	12.9	

OR: Odds ratio; FSFI: Female Sexual Function Index; PCOS: Polycystic ovary syndrome; NS: not signified.

## Data Availability

No new data were created or analyzed in this study.

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
