# Peer review of "Sexual Dysfunction in Patients with Chronic Non-Genital Physical Disease: An Umbrella Review"

_ijerph, 2025, doi:10.3390/ijerph22020157_

Round 1
Reviewer 1 Report
Comments and Suggestions for Authors
General comment
The manuscript entitled “Sexual Dysfunction in Patients with Chronic Non-Genital Physical Disease: An Umbrella Review” is a valuable and comprehensive umbrella review addressing the understudied topic of sexual dysfunction in patients with chronic non-genital physical diseases. It effectively synthesizes existing systematic reviews and meta-analyses to provide insights into the prevalence and impact of these dysfunctions across various conditions. The adherence to PRISMA guidelines and pre-registration in PROSPERO enhances its methodological rigor. Few corrections are suggested to improve the quality of the work:
- The introduction effectively sets the stage, but it could benefit from a more in-depth discussion of why non-genital diseases have been historically overlooked in sexual health research.
- The methodology is robust; however, including search equations in the supplementary material or main text would enhance reproducibility.
- While comprehensive, the presentation could be improved by condensing disease-specific findings into summary tables or infographics for clarity.
- About comorbidities that could impact sexual health also see: DOI: 10.1016/j.dld.2022.05.016 , DOI: 10.1111/jop.13362 and DOI: 10.1111/and.14504
- The section provides insightful interpretations, but a deeper exploration of cultural and societal factors influencing sexual dysfunction in chronic disease patients would add value.
- The call for more female-centered research is commendable, but specific recommendations for improving clinical practice and training could be more explicit.
Author Response
Comment :
1) The introduction effectively sets the stage, but it could benefit from a more in-depth discussion of why non-genital diseases have been historically overlooked in sexual health research.
- We thank the reviewer for this comment and have added the following sentences to the introduction to focus on the lack of sexuality in the study of chronic non-genital diseases:
Sexuality has long been studied primarily from the perspective of reproduction, with attention focused on the genital organs and reproductive functions. This perspective has led to a reductive view of sexuality, limited to its genital aspects.
2) The methodology is robust; however, including search equations in the supplementary material or main text would enhance reproducibility.
- We are happy to read that the reviewer considers the methodology as robust. The search equations consist of 86 pages in Word format, written in Arial 11. This is too long to include them in the text, but are made available to the readers as supplementary material, along with the tables, figures, and notes. A phrase has been added to the article mentioning explicitly that the search equations can be provided upon request.
3) While comprehensive, the presentation could be improved by condensing disease-specific findings into summary tables or infographics for clarity.
- We thank the reviewer for this comment. A table was supplied with the previous version as an annex to the submitted paper, but it appear this was not available to the reviewers. It is now provided in the text.
4 ) About comorbidities that could impact sexual health also see: DOI: 10.1016/j.dld.2022.05.016 , DOI: 10.1111/jop.13362 and DOI: 10.1111/and.14504
- We are grateful for these reference. The enables us to add a small paragraph on comorbities to the text, stating: Mood disorders, particularly anxiety and depression, emerge as significant comorbidities that intensify sexual dysfunction. In gastrointestinal disorders, stress and anxiety induced by the underlying pathology play a critical role in the development of sexual dysfunction, emphasizing the psychological burden these conditions impose on sexual health [79]. Similarly, in men with obstructive sleep apnoea syndrome (OSAS), chronic fatigue and sleep disturbances directly affect both mental and sexual health, worsening symptoms of erectile dysfunction [80].
5) The section provides insightful interpretations, but a deeper exploration of cultural and societal factors influencing sexual dysfunction in chronic disease patients would add value.
-
We have added a line on this subject in the paragraph on perspectives for the future: Additionally, it is crucial to consider the influence of societal and cultural factors, such as the stigmatization of sexuality, religious norms, lack of sex education, and age-related stereotypes, which often hinder the recognition and management of sexual dysfunction in patients with chronic illnesses
6) The call for more female-centered research is commendable, but specific recommendations for improving clinical practice and training could be more explicit.
- We thank the reviewer for this suggestion and have added the following sentence: In this context, it seems that doctors could play a more active role by systematically including sexual health in the clinical assessment of chronic illnesses. Similarly, encouraging patients to discuss these issues more openly could lead to better care. A two-way approach, combining greater responsibility on the part of carers and greater involvement on the part of patients, could thus help to improve the management of sexual dysfunction.

Reviewer 2 Report
Comments and Suggestions for Authors
- The article addresses important aspects of sexual dysfunction in individuals suffering from chronic non-genital diseases through an umbrella review. It identifies the CD, various aspects of sexual dysfunction in both genders and provides a comprehensive account of available evidence in literature. It encompasses a wide range of CDs. It reports significant aspects of SD in CDs, and highlights the need for screening in individuals with CDs. It will be of value to those practicing in the field.
- Abstract is structured and is representative of the study.
- The title is appropriate and the introduction adequately provides an overview of existing literature, identifies the gap, and states the goals of the study.
- Methodology is elaborate, well-described, and clear. A detailed explanation of the procedure, outcome measures, and statistical analyses are given. However, it could not be assessed fully because tables are neither included in the manuscript nor provided as supplementary material for the reviewers. Therefore, the reproducibility of the study could not be assessed.
- Result is presented in detail. However, the figures and tables, as stated earlier, were not available for this reviewer to assess.
- The discussion explores the results in detail and compares them with previous evidence. It further mentions the limitations. It is suggested to change the sub-heading study limitations to “strengths and limitations”.
- The conclusion section is consistent with the results and interpretation.
- Minor modifications-
o CD has been expanded in the first mention in the manuscript. All subsequent mentions can be replaced with the abbreviation.
o The author contribution section needs to be filled.
- The references are appropriate, follow a consistent style, and well-cited.

Author Response
Comment :
1) However, the figures and tables, as stated earlier, were not available for this reviewer to assess.
- We regret that you did not have access to the tables mentioned in the text for the previous version. To facilitate the review process, we have now included the tables in the text, rather than as an annexe
2) It is suggested to change the sub-heading study limitations to “strengths and limitations”.
- We thank the reviewer for this suggestion and have changed the subtitle accordingly.
3) CD has been expanded in the first mention in the manuscript. All subsequent mentions can be replaced with the abbreviation.
- We thank the reviewer for this suggestion and have changed the manuscript accordingly.
4) The author contribution section needs to be filled.
- We have added the following specification: Author Contributions: Study concept: CL and SV; methodology: CL; software (not applicable); validation: CL and CJ; database search, analysis and data curation, CL and CJ; resources (not applicable); writing—original draft preparation: CL and SV; writing—review and editing: CL; visualization: CL; supervision: SV; project administration: CL; funding: not applicable. All authors have read and agreed to the published version of the manuscript.

Reviewer 3 Report
Comments and Suggestions for Authors
I reviewed the paper entitled Sexual dysfunction in patients with chronic non-genital physi-2 cal disease: An umbrella review .the paper is very good in topic and interesting basic information this review could be used as a future reference for this importantly stated topic. But many recommendation should be stated
Abstract lines 26-35 should be reorganized again as there are in bad sequence must be recognized
Keywords must be reorganized with alphabet order
Introduction is very good but second paragraph need more references to be more informative and clear
Last paragraph in introduction must be reorganized again
Materials and methods
Eligibility criteria Authors stated this criteria but what about exclusion criteria as well as limitation?
Data exctraction this paragraph I could not understand it so much please rephrasing is needed
Gastroenterological diseases and cancer paragraphs should be minimized and reduced please make this change many information present not related to this topic
Thank you again to let me reviewed this paper
Author Response
Thank you very much for taking the time to review this manuscript.
1) Abstract lines 26-35 should be reorganized again as there are in bad sequence must be recognized
- We thank the reviewer for this comment. We have adapted the text as follows: Results: Among men, CD is associated with erectile dysfunction, and among females, with lower desire, arousal, lubrication, orgasm and sexual satisfaction, and with increased pain during intercourse. For both men and women, depression, anxiety and fatigue are also reported, while women with CD are more affected by a poor body image than men. Clinical implications: Patients with CD, especially females, should be more routinely assessed for the impact of their condition on sexual functioning. The impact of CD on men’s sexuality has been extensively studied in terms of erectile capacity, but other aspects of their sexuality are largely neglected. Strengths and limitations: This is the first umbrella review to bring together the documented findings regarding sexual dysfunction among patients with various non-genital CD. While the heterogeneity of the CD makes the study unique and clinically relevant, it renders the interpretation of the results more difficult. The overrepresentation of men in existing studies reflects the current state of research but limits the applicability of the findings for women. Conclusion: Women and men with non-genital CD can suffer from SD or reduced sexual function. Health professionals should pay more attention to managing these sexual disorders, even when the disease does not affect the genitals.
2) Keywords must be reorganized with alphabet order
-
Thank you for the suggestion, we have changed the order of the keywords Keywords:chronic disease; sexual dysfunction; umbrella review
3) Introduction is very good but second paragraph need more references to be more informative and clear.
-
We thank the reviewer for this suggestion and have adapted the text as follows:
Chronic diseases, defined as long-term conditions lasting at least six months and evolving over time, represent a significant global health burden, accounting for approximately 17 million deaths annually [1]. The Community of Patients for Research [7] classifies CD into 15 categories: (1) cardiovascular diseases, (2) cancers, (3) endocrine diseases, (4) respiratory and ear, nose, and throat (ENT) diseases, (5) digestive system diseases, (6) rheumatological diseases, (7) neurological and muscular diseases, (8) psychiatric and psychological diseases, (9) renal, urinary and genital diseases, (10) skin diseases, (11) eye diseases, (12) systemic diseases, (13) chronic infectious diseases, (14) hematological diseases, and (15) neurodiversities. This classification is also recognized in the list of CD adapted from the International Classification of Primary Care, drawn up by the International Organization of General Practitioners [8] and the International Classification of Diseases, which provides a common language for CD [9]. Such classifications are critical for understanding the diverse health impacts of chronic diseases, including their influence on sexual functioning and overall quality of life.
4) Eligibility criteria Authors stated this criteria but what about exclusion criteria as well as limitation?
- We thank the reviewer for this comment and have provided more detailed explanations for the reasons for exclusion: Other conditions excluded from the study were paraphilic disorders and gender identity disorders and chronic malformations occurring after surgery (which are no longer listed as disorders). Similarly, chronic mental disorders of which the management is very different from that of chronic physical diseases. The sexual repercussions of aging and menopause were also excluded, as these are not considered chronic diseases. Aging and menopause are natural physiological processes and part of the normal course of life. While they can significantly impact sexuality, they do not meet the criteria for chronic diseases, which typically involve an underlying pathology or medical disorder requiring specific management. Depression and anxiety are also not considered as chronic diseases included in the study, because they are either chronic mental illness, which falls under the exclusion criteria, or comorbidities of other CD. In terms of populations, studies on children or adolescents were excluded from the review, as were studies involving animal testing because the links with sexuality cannot be established in any meaningful way in these contexts (3). For the purposes of this umbrella review, narrative reviews, case studies, primary reviews, clinical trials, cohort studies, case-control studies and randomised controlled trials were excluded. (4) In addition, reviews not published in peer-reviewed journals, such as doctoral theses and conference presentations or posters, were also excluded.
- The reasons for exclusion are also highlighted in the flowchart.
5) Data exctraction this paragraph I could not understand it so much please rephrasing is needed
-
We have changed the text as follow for greater clarity:
The full-text versions of all publications meeting the inclusion criteria were reviewed by two independent researchers (CL, CJ). They extracted the following information:
- Study characteristics: authors, year of publication, title, study design, number of reviewers, databases consulted, keywords, inclusion and exclusion criteria, quality assessment, and diagnostic tool used for identifying sexual dysfunction.
- Population characteristics: gender, type of chronic disease, country of the study, and sample size.
- Clinical findings: presence of sexual dysfunction, counseling provided, study limitations, and main conclusions.
The data extracted from the included studies were summarized in a Microsoft Excel spreadsheet. Any disagreements or discrepancies were resolved through consensus or, if necessary, by consulting a third investigator (SV).
6) Gastroenterological diseases and cancer paragraphs should be minimized and reduced please make this change many information present not related to this topic
- We thank the reviewer for this suggestion and have reduced the text as much as possible. The sentences have been crossed out in the revised version of the manuscript.
Reviewer 4 Report
Comments and Suggestions for Authors
Dear authors, after reading your paper, in my opinion, there are some drawbacks:
1. There are some limited justification for limitations. While the text lists excluded conditions, it doesn’t always provide detailed reasoning. For example, excluding studies on "old age and menopause" could benefit from an explanation of why these do not align with the review's goals.
2. The PRISMA flowchart is mentioned but not included.
3. Some details, such as the limitations of studies (e.g., absence of control groups, critically low quality), are repeated multiple times, reducing readability and conciseness.
4. While individual diseases are discussed, there is minimal cross-comparison between conditions, tools, or gender differences. For example, the impact of cardiovascular disease on sexual functioning could be compared to that of cancer to provide broader insights.
5. There is a limited insight on gender. Gender differences are highlighted in some areas (e.g., stroke, cancer), but the discussion often lacks depth. For instance, the text mentions differences in the prevalence and types of sexual dysfunctions but does not explore underlying reasons or implications.
6. The absence of control groups is frequently noted but not discussed in terms of how it affects the reliability or interpretability of the findings.
7. The tables you are mentioning in the text are missing
Author Response
Thank you very much for taking the time to review this manuscript.
1) There are some limited justification for limitations. While the text lists excluded conditions, it doesn’t always provide detailed reasoning. For example, excluding studies on "old age and menopause" could benefit from an explanation of why these do not align with the review's goals.
-
We thank the reviewer for these comments, and have adapted the text as follows, to further justify the choice of exclusion criteria.
The following publications were not included (exclusion criteria): (1) Publications focusing on chronic genital gynecological disorders such as endometriosis, uterine fibroids, ovarian cysts, and fertility disorders, or on chronic genital urological disorders such as benign prostatic hypertrophy and adenoma, bladder leakage, overactive bladder, or primary erectile dysfunction. Also excluded were studies related to cancers of the genitals such as bladder, prostate, breast, cervical, vaginal, penile, and testicular. As these have an obvious impact on sexual functioning, they have already been studied extensively, and as such do not concur with the focus and objective of the present review. Although the breast is not really a sexual organ, studies on the effects of breast cancer on sexual functioning were also excluded for the same reason. Conversely, studies on anal cancer were included, as the anus is not considered a genital area. Other conditions excluded from the study were paraphilic disorders and gender identity disorders, as well as chronic malformations occurring after surgery (which are no longer listed as disorders), and chronic mental disorders, the management of which is very different from that of chronic physical diseases. The sexual repercussions of aging and menopause were also excluded, as these are not considered chronic diseases. Aging and menopause are natural physiological processes and part of the normal course of life. While they can significantly impact sexuality, they do not meet the criteria for chronic diseases, which typically involve an underlying pathology or medical disorder requiring specific management. Depression and anxiety are not considered as chronic diseases included in the study, because they are either chronic mental illness, which falls under the exclusion criteria, or comorbidities of other CD. In terms of populations, studies on children or adolescents were excluded from the review, as were studies involving animal testing because the links with sexuality cannot be established in any meaningful way in these contexts. (3) For the purposes of this umbrella review, narrative reviews, case studies, primary reviews, clinical trials, cohort studies, case-control studies and randomised controlled trials were excluded. (4) In addition, reviews not published in peer-reviewed journals, such as doctoral theses and conference presentations or posters, were also excluded.
2. The PRISMA flowchart is mentioned but not included.
- We apologise for the error. The PRISMA table was supplied with the previous version (as were the other tables), but in a separate document. We have included it in the revised manuscript to facilitate the review.
3. Some details, such as the limitations of studies (e.g., absence of control groups, critically low quality), are repeated multiple times, reducing readability and conciseness.
- We agree with the reviewer that some elements are repeated. To increase readability and conciseness, we have reformulated parts of the text that were repetitive and adapted it to enhance readability. The changes made to the text to that effect are in red and deleted sentences made visible. .
4. While individual diseases are discussed, there is minimal cross-comparison between conditions, tools, or gender differences. For example, the impact of cardiovascular disease on sexual functioning could be compared to that of cancer to provide broader insights.
- We thank the reviewer for raising this important point, and agree that adding comparisons would add to the review. While the main objective was to assess the extent to which non-genital chronic diseases affect sexual functioning, it would have been interesting to consider quantitative differences in severity by disease or by sex. However, as this is a review paper we were restrained by the methodology used in the papers that were included. Only a small number of the studies involved a comparative aspect, and those that did used heterogeneous measures. The use of odds ratios, relative risks and prevalence rates provides for an overall overview, but are too general to allow an inter-comparison. In the discussion, we note this as a limitation of the study. Nevertheless, to accommodate partly to this limitation, we present an overview of the odds ratios for the effects of different chronic diseases allowing some form of comparison. In addition, the main finding that all of the non-genital chronic diseases considered in this review affect sexual functioning remains valuable. While we agree that a visual representation of the effects would be very interesting, this form of “meta-meta-analysis” is not feasible for an umbrella review involving reviews that aggregate the results from a variety of primary studies, and in many cases do not provide the necessary information to allow such calculations.
5. There is a limited insight on gender. Gender differences are highlighted in some areas (e.g., stroke, cancer), but the discussion often lacks depth. For instance, the text mentions differences in the prevalence and types of sexual dysfunctions but does not explore underlying reasons or implications.
- We thank the reviewer for this comment. While our manuscript can only synthesize the findings from existing reviews, hardly any of which investigated underlying reasons for gender differences, it was not possible to analyse these and present them in the results. However, we elaborated this point in the discussion section. A notable point in this review is the striking finding that female sexuality is under-represented in reviews and meta-analyses, compared with studies of men. We have emphasized this point in the discussion section in the form of hypotheses for further research and suggest that more in-depth research is necessary to gain a better understanding of this disparity Specifically, we have added the following: The results of this review highlight a gender disparity, with women being under-represented in studies of the impact of chronic diseases on sexuality. This under-representation has led to a downplaying of women's health issues, including their sexual health, which is considered a lower priority in studies of human sexuality [76]. On the other hand, female sexuality is typically perceived as more complex or difficult to assess, particularly when CD is accompanied by depression or mood disorders, which may discourage researchers from investigating this area in depth [77]. This contributes to a lesser understanding of the interactions between chronic diseases and sexuality in women, despite the importance of these issues for their quality of life and overall well-being. Further research is needed to explore the reasons for the gender differences that are observed in the impact of CD on men’s and women's sexuality.
6. The absence of control groups is frequently noted but not discussed in terms of how it affects the reliability or interpretability of the findings.
- Thank you to the reviewer for this comment. Indeed, the absence of control groups in most of the studies is an observation whose impact on the validity of the results can be highlighted in the discussion. We have adapted the discussion by making the following changes. Finally, it is important to note that many of the studies included in this review did not include control groups, which limits the ability to determine whether the reported effects differ significantly from natural trends in sexual functioning in the general population, particularly the increasing incidence of sexual problems with age. Consequently, the conclusions should be interpreted with some caution. In addition, chronic illnesses, including multimorbidity, are particularly common, especially in elderly populations, making it difficult to create perfectly comparable control groups. These methodological limitations must be taken into account, and underline the need for future research incorporating more rigorous approaches to explore these interactions more precisely.
7. The tables you are mentioning in the text are missing
- We apologise for the error. The tables were provided with the previous version (as were the other tables), but in separate documents. We have included them in the revised manuscript to facilitate the review.

Round 2
Reviewer 1 Report
Comments and Suggestions for Authors
The authors improved the manuscript according to previous suggestions. No further corrections required.
Reviewer 4 Report
Comments and Suggestions for Authors
Dear authors, after reading the revised version of the article, I think you have made the necessary modifications and the editorial process can continue.